

# Impact of nonalcoholic fatty liver disease on the functional cure of nucleos(t)ide analogues-treated chronic hepatitis B patients add-on pegylated interferon therapy: a retrospective study

Huili Li, Ling Li, Yiru Zhao, Guangde Yang, Xia Wang, Juanjuan Fu, Li Li and Xiucheng Pan

The Affiliated Hospital of Xuzhou Medical University, Xuzhou Medical College, Jiangsu, Xuzhou, China

## ABSTRACT

**Background**. This study aimed to determine whether nonalcoholic fatty liver disease (NAFLD) influences the functional cure of nucleos(t)ide analogues (NAs)-treated chronic hepatitis B (CHB) patients in combination with pegylated interferon (PEG-IFN) therapy.

**Methods**. A retrospective analysis was conducted on CHB patients treated with nucleos(t)ide analogues (NAs) who received PEG-IFN combination therapy at a single center. Patients were stratified into a CHB group and a CHB-NAFLD group based on the presence of fatty liver comorbidity. The primary endpoint was the difference in hepatitis B virus surface antigen (HBsAg) seroclearance rates between the two groups, and the secondary endpoint was the differences in biochemical parameters and adverse events.

**Results**. A total of 158 NAs-treated CHB patients were enrolled, comprising 91 CHB-only patients and 67 CHB-NAFLD patients. After 48 weeks of PEG-IFN therapy, 20.9% of patients achieved HBsAg seroclearance. Before and after propensity score matching (PSM), differences in cumulative HBsAg seroclearance probability between groups were non-significant (25.3% vs. 14.9%, $P = 0.121$; 32.4% vs. 13.5%, $P = 0.063$). Similarly, cumulative hepatitis B e antigen (HBeAg) seroconversion incidence showed no intergroup differences before and after PSM ($P = 0.618$ and $P = 0.954$, respectively). Logistic regression analysis indicated that baseline HBeAg status was independently associated with HBsAg seroclearance. Further analysis confirmed that NAFLD did not significantly affect HBsAg loss, regardless of HBeAg status, both before and after PSM ($P > 0.05$). Moreover, the NAFLD group exhibited higher rates of abnormal alanine aminotransferase (ALT) at weeks 24 and 48 versus the CHB-only group (31.9% vs. 12.3%, $P = 0.009$; 70.8% vs. 47.4%, $P = 0.007$). However, ALT normalization rates 24 weeks post-treatment showed no significant difference (88.9% vs. 77.2%, $P = 0.074$).

**Conclusion**. NAFLD does not affect the achievement of functional cure with PEG-IFN therapy in NAs-treated chronic hepatitis B patients. However, CHB patients with NAFLD show a reduced likelihood of ALT normalization during treatment.

Corresponding authors
Li Li, three125@sina.com
Xiucheng Pan, xzpxc68@126.com

# INTRODUCTION

Despite widespread vaccine and antiviral use significantly reducing new hepatitis B virus (HBV) infections, the global burden of chronic HBV infection remains substantial. According to the World Health Organization (WHO) 2021 data, approximately 296 million people worldwide had chronic HBV infections, resulting in about 887,000 annual deaths from HBV-related complications including liver failure, cirrhosis, hepatocellular carcinoma, and other associated conditions (*Hsu, Huang & Nguyen, 2023*; *Singal, Kanwal & Llovet, 2023*). These statistics demonstrate that chronic hepatitis B (CHB) remains a major challenge in global public health.

Nonalcoholic fatty liver disease (NAFLD) refers to a spectrum of fatty liver disorders that occur in the absence of secondary causes of hepatic steatosis. These secondary causes include alcohol consumption, hepatitis C, parenteral nutrition, steatogenic medications, and inherited metabolic disorders (*Chalasani et al., 2018*). It is reported that NAFLD is often associated with metabolic comorbidities such as obesity, diabetes, and dyslipidemia in the majority of patients (*Chalasani et al., 2018*) . In recent years, lifestyle and dietary changes have contributed to rising rates of obesity and metabolic syndrome. Consequently, NAFLD has emerged as a prevalent chronic liver disease, affecting approximately 24% of the global population (*Powell, Wong & Rinella, 2021*). With increasing NAFLD prevalence, NAFLD-CHB comorbidity has become a significant clinical feature with growing epidemiological importance. The overall prevalence rate of NAFLD among CHB patients is approximately 29.0% (range: 13.5%–56.0%) (*Yang & Wei, 2022*). Therefore, investigating the interaction between CHB and NAFLD as well as its impact on the prognosis of the diseases has become a major research focus. Several studies suggest that concurrent NAFLD may affect the natural course and outcome of CHB, for example, *Huang et al. (2024)* reported positive correlations between NAFLD and both HBV-DNA suppression and hepatitis B virus surface antigen (HBsAg) seroclearance in untreated CHB patients (*Wang et al., 2014*).

However, the impact of NAFLD on antiviral treatment efficacy for CHB remains controversial. Some studies suggest that NAFLD may compromise the therapeutic benefits of antiviral therapy in CHB, while others demonstrate no significant correlation between hepatic steatosis in CHB patients and complete remission post-antiviral treatment (*Shi et al., 2012*; *Tang et al., 2023*).

Both CHB and NAFLD can lead to severe liver-related complications. Evidence indicates that CHB patients with hepatic steatosis have a significantly increased risk of liver fibrosis progression and cancer development (*Mak et al., 2020*; *Peleg et al., 2019*; *Wong et al., 2014*). Effective antiviral therapy is essential to prevent CHB progression and improve long-term prognosis, while achieving functional cure substantially reduces risks of liver cancer and other adverse outcomes . Current guidelines recommend two major classes of anti-HBV drugs: nucleos(t)ide analogues (NAs) and pegylated interferon (PEG-IFN) (*Tao et al., 2020*).

NAs can effectively inhibit HBV replication but require long-term administration, with only a minority of patients achieving HBsAg clearance (*Chevaliez et al., 2013*; *Mo et al., 2024*). PEG-IFN possesses dual immunomodulatory and antiviral effects, which

can dramatically improve the HBsAg clearance rate in CHB patients (3%–7%) (*Hu et al., 2018*). Furthermore, multiple studies demonstrate that combination or sequential PEG-IFN therapy in NAs-treated CHB patients can achieve clearance rates exceeding 20% with enhanced response durability (*Hu et al., 2018*; *Mo et al., 2024*; *Ning et al., 2014*). Although evidence suggests NAFLD may suppress HBV replication, its impact on functional cure attainment in NAs-treated CHB patients undergoing PEG-IFN therapy remains unclear and warrant further investigation.

Therefore, the primary purpose of this study is to assess the impact of NAFLD on achieving HBsAg seroclearance in NAs-treated CHB patients undergoing PEG-IFN combination therapy.

## MATERIALS & METHODS

### Patient

This retrospective study analyzed clinical data from NAs-treated CHB patients who received Peg-IFN at the Affiliated Hospital of Xuzhou Medical University between September 1, 2018 and December 31, 2022 (Fig. 1). Inclusion criteria: (1) age ≥18 years; (2) CHB diagnosis according to the "*Chinese guidelines for the prevention and treatment of chronic hepatitis B (version 2022)*"; (3) NAs treatment duration ≥1 year; (4) levels of HBV DNA <20 IU/mL; (5) available liver ultrasound or FibroScan or other abdominal imaging reports before initial interferon treatment and at the end of follow-up. Exclusion criteria: (1) co-infection with other viruses (such as hepatitis C virus, hepatitis D virus, human immunodeficiency virus); (2)concurrent alcoholic hepatitis, autoimmune liver disease, decompensated cirrhosis, or malignancies; (3)pregnant or lactating women; (4) lack of baseline or follow-up data.

### Study design and assessment

Patients were divided into the CHB group and NAFLD-CHB group based on the presence of NAFLD at baseline. All NAs-treated patients received combined PEG-IFN treatment. The NAs mainly included Entecavir (Jiangxi Pharmaceutical Co, Ltd., China), Tenofovir disoproxil fumarate (Qilu Pharmaceutical Co, Ltd., Shandong, China), Tenofovir alafenamide Fumarate tablets (TAF; Qilu Pharmaceutical Co, Ltd., Shandong, China; Thermo Fisher Scientific, Waltham, MA, USA), Tenofovir Amibufenamide (JIANGSU HANSOH PHARMA, Jiangsu, China). PEG-IFN-α (Xiamen Amoytop Biotech Co, LTD., Xiamen, Fujian, China) was administered subcutaneously at a dose of 180 μg weekly for 48 weeks. The baseline was defined as the time of the first administration of PEG-IFN-α, and the follow-up continued until 24 weeks after the completion of PEG-IFN treatment. The primary objective of the study was to compare the difference in HBsAg clearance between the CHB group and the CHB-NAFLD group, and the secondary objective was to analyze the distinction in HBsAg decline, hepatitis B e antigen (HBeAg) clearance, biochemical response, and adverse reactions between the two groups.

 

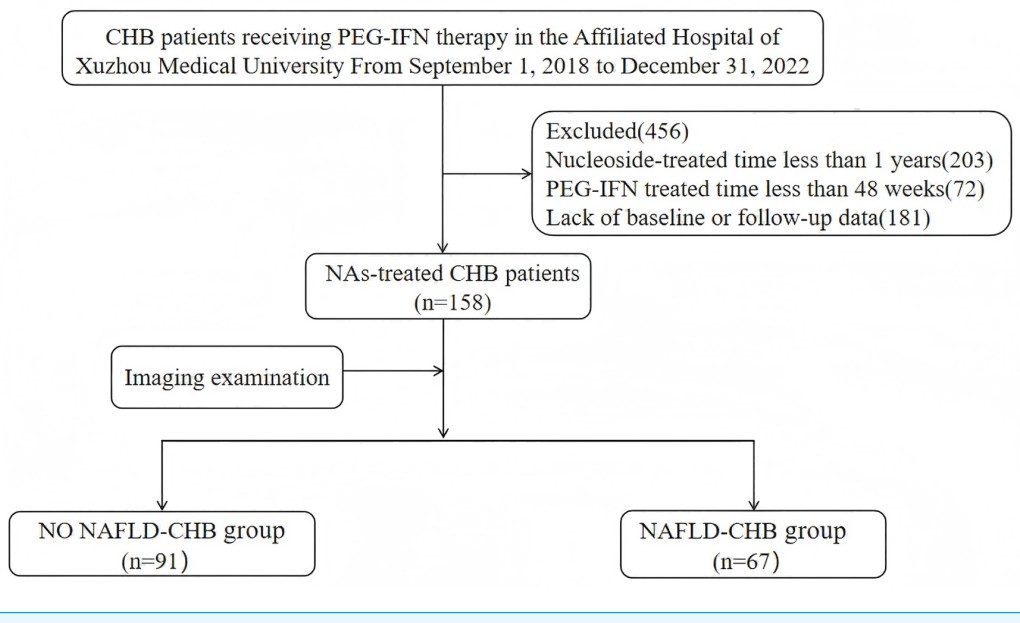

**Figure 1   The patients flowchart.**

## Data collection

Patient clinical data were extracted from the electronic medical record system of Xuzhou Medical University Affiliated Hospital. Data related to patient age, sex, family history of liver disease, NAs type and treatment duration, adverse reactions, HBV DNA, HBV serum infection markers, complete blood counts, blood biochemistry, and imaging were collected. Clinical follow-up data were updated quarterly through outpatient visits and telephone interviews.

## Statistical analysis

The statistical analysis was conducted using SPSS 27.0 software. Normally distributed continuous variables are presented as mean $\pm$ standard deviation, and compared using independent samples $t$-tests. For non-normally distributed continuous variables, data are expressed as the median (with 25th and 75th percentiles), and compared with Mann–Whitney $U$ tests. Categorical data are expressed as percentages (%) and analyzed by the chi-square test. When the total sample size n < 40 or when the expected frequency in any cell was <5, the categorical data should be compared using Fisher's exact test. Binary logistic regression was used to identify factors associated with HBsAg seroclearance. Propensity score matching (PSM) was performed to overcome baseline imbalances. The Kaplan–Meier analysis compared cumulative HBsAg seroclearance incidence between groups, with survival curves assessed by log-rank tests. $P$ value < 0.05 was considered to indicate statistical significance.

**Table 1  Baseline characteristics of patients before and after propensity matching.**

| Variable | Before PSM | | | After PSM | | |
|---|---|---|---|---|---|---|
| | CHB without NAFLD ($n = 91$) | CHB with NAFLD ($n = 67$) | P | CHB without NAFLD ($n = 37$) | CHB with NAFLD ($n = 37$) | P |
| AGE, years | $40.0 \pm 8$ | $39.0 \pm 8$ | 0.616 | $39.0 \pm 7$ | $39.0 \pm 8$ | 0.933 |
| SEX, n (%) | | | 0.209 | | | 0.711 |
| Male | 75 (82.4) | 60 (89.6) | | 34 (91.9) | 32 (86.5) | |
| Female | 16 (17.6) | 7 (10.5) | | 3 (8.1) | 5 (13.5) | |
| Family history n (%) | 15 (16.5) | 15 (22.4) | 0.350 | 7 (18.9) | 9 (24.3) | 0.572 |
| NAs-treated time years | 4 (2.0–6.0) | 5 (2.0–8.0) | 0.055 | 4 (2.5–6.5) | 4 (2.0–7.0) | 0.857 |
| HBsAg, IU/mL | 619.6 (137.6–1,152.1) | 605.9 (208.9–1,294.1) | 0.581 | 435.0 (161.4–1,271.6) | 696.2 (177.9–1,243.3) | 0.499 |
| HBsAg, n (%) | | | | | | |
| <100 IU/mL | 19 (20.9) | 11 (16.4) | 0.480 | 6 (16.2) | 6 (16.2) | 1.000 |
| 100–500 IU/mL | 22 (24.2) | 21 (31.3) | 0.317 | 13 (35.1) | 8 (21.6) | 0.197 |
| 500–1,500 IU/mL | 38 (41.8) | 23 (34.3) | 0.343 | 14 (37.8) | 19 (51.4) | 0.242 |
| ≥1,500 IU/mL | 12 (13.2) | 12 (17.9) | 0.414 | 4 (10.8) | 4 (10.8) | 1.000 |
| HBeAg, IU/mL | 0.32 (0.03–4.77) | 0.39 (0.03–11.27) | 0.539 | 0.75 (0.07–13.47) | 0.40 (0.02–12.68) | 0.545 |
| HBeAg (+), n (%) | 34 (37.4) | 27 (40.3) | 0.708 | 16 (43.2) | 16 (43.2) | 1.000 |
| ALT, U/L | 20.0 (14.0–29.0) | 28.0 (19.0–40.0) | 0.002 | 22.0 (14.0–32.0) | 24.0 (17.0–30.0) | 0.820 |
| AST, U/L | 20.0 (16.0–24.0) | 21.0 (18.0–26.0) | 0.165 | 18.0 (16.0–25.0) | 20.0 (18.0–24.0) | 0.807 |
| GGT, U/L | 18.0 (13.0–25.0) | 30.0 (19.0–40.0) | <0.001 | 24.0 (17.0–34.0) | 20.0 (17.0–31.0) | 0.638 |
| ALP, U/L | 70.0 (60.0–79.0) | 74.0 (60.0–85.0) | 0.227 | 70.0 (59.0–82.0) | 74.0 (61.0–85.0) | 0.402 |
| TP, g/L | $75.8 \pm 3.8$ | $76.4 \pm 4.2$ | 0.353 | $76.5 \pm 4.5$ | $77.0 \pm 3.4$ | 0.596 |
| ALB, g/L | $48.4 \pm 2.4$ | $48.9 \pm 2.7$ | 0.223 | $49.2 \pm 2.4$ | $49.2 \pm 2.2$ | 0.904 |
| TBIL, μmol/L | 13.0 (8.6–16.8) | 12.5 (9.6–15.5) | 0.915 | 12.4 (8.8–17.5) | 11.9 (9.5–16.2) | 0.779 |
| DBIL, μmol/L | 4.6 (3.5–6.0) | 4.6 (3.6–5.5) | 0.675 | 4.6 (3.5–6.2) | 4.2 (3.5–5.3) | 0.417 |
| AFP, ng/mL | 3.3 (2.2–4.5) | 2.8 (2.3–3.4) | 0.046 | 3.1 (2.1–4.7) | 2.8 (2.3–3.8) | 0.486 |
| UREA, mmol/L | $4.6 \pm 1.0$ | $5.1 \pm 1.0$ | 0.003 | $4.8 \pm 1.0$ | $5.0 \pm 1.1$ | 0.522 |
| CREA, μmol/L | $66.5 \pm 12.3$ | $70.3 \pm 12.7$ | 0.059 | $68.9 \pm 10.5$ | $69.1 \pm 13.1$ | 0.961 |
| UA, μmol/L | $322.3 \pm 64.5$ | $346.7 \pm 71.5$ | 0.026 | $344.2 \pm 58.1$ | $325.9 \pm 71.8$ | 0.232 |
| GLU, mmol/L | 5.7 (5.2–5.9) | 5.7 (5.1–6.0) | 0.975 | 5.6 (4.8–6.9) | 6.4 (5.5–7.1) | 0.975 |
| WBC, $\times 10^9$/L | 5.6 (4.7–6.4) | 6.4 (5.3–7.1) | 0.006 | 5.6 (4.8–6.9) | 6.4 (5.5–7.1) | 0.146 |
| NLR | 1.8 (1.4–2.3) | 1.7 (1.2–2.2) | 0.367 | 1.8 (1.4–2.2) | 1.7 (1.3–2.1) | 0.534 |
| PLT, $\times 10^9$/L | $202.1 \pm 51.0$ | $227.2 \pm 48.7$ | 0.002 | $213.4 \pm 54.0$ | $216.1 \pm 50.1$ | 0.827 |

**Notes.**

ALT, alanine aminotransferase; AST, aspartate aminotransferase; GGT, gamma-glutamyl transferase; ALP, alkaline phosphatase; TP, total protein; ALB, albumin; TBIL, total bilirubin; DBIL, direct bilirubin; AFP, alpha fetoprotein; UREA, urea; CREA, creatinine; UA, Uric Acid; GLU, blood glucose; WBC, white blood cell; NLR, neutrophil-to-lymphocyte ratio; PLT, platelet.

## Research ethics

The study protocol was reviewed and approved by the Affiliated Hospital of Xuzhou Medical University Medical Ethics Committee (Ethics number: XYFY2024-KL225-01). Informed consent was not required because the study design was retrospective.

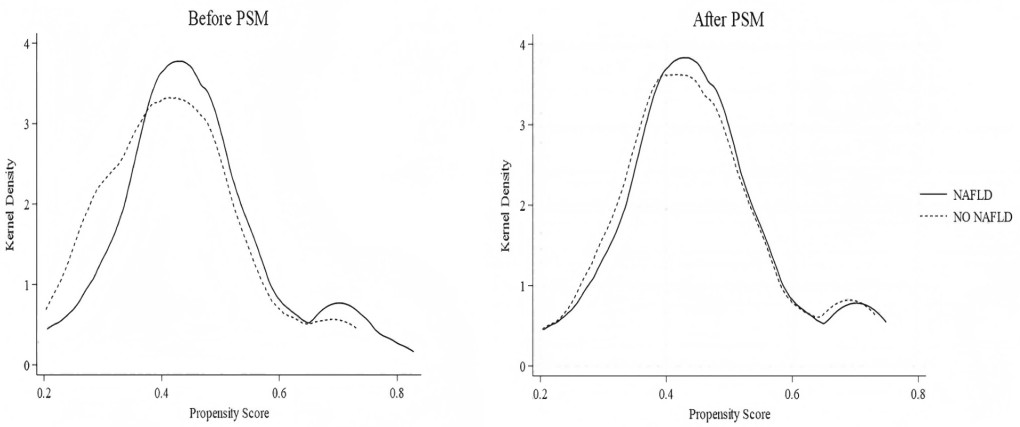

**Figure 2** Kernel density plots of propensity scores before and after matching.

## RESULTS

### Baseline characteristics of patients

A total of 158 NAs-treated CHB patients receiving PEG-IFN treatment were enrolled in the study, comprising 91 in the CHB-only group and 67 in the NAFLD-CHB group. The baseline characteristics of the enrolled patients are presented in Table 1. The mean age of these patients was 39.0 ± 8.0 years, and 85.4% were male. At baseline, 61 patients (38.6%) were HBeAg-positive, median NAs duration was 4.0 years (IQR 2.0–6.0), and HBsAg levels were comparable between groups. The NAFLD-CHB group demonstrated significantly higher alanine aminotransferase (ALT), gamma-glutamyl transferase (GGT), white blood cell (WBC) count, and platelet (PLT) count compared to the CHB group ($P < 0.05$). To minimize the potential impact of confounding factors, these variables were incorporated into propensity score matching with a caliper width of 0.02. After successful matching of 37 pairs, baseline characteristics were balanced with no significant intergroup differences.The detailed results are summarized in Table 1. The matching results were assessed for robustness using kernel density distribution plots. These plots demonstrate overlapping distributions of propensity scores between the two groups after PSM, suggesting improved balance, as shown in Fig. 2.

### The effect of NAFLD on HBsAg during PEG-IFN therapy in patients with CHB

During PEG-IFN treatment, both groups exhibited declining HBsAg trends from baseline. Intergroup comparisons of HBsAg levels at each follow-up time point, both before and after matching, there were no significant differences were observed (Figs. 3A and 3B). Furthermore, the magnitude of HBsAg reduction from baseline at weeks 12, 24, 48, and 72 showed no statistically significant differences between groups before or after PSM ($P > 0.05$). At the end of the follow-up, 33 patients (20.9%) achieved HBsAg seroclearance. The CHB-only group showed higher seroclearance rates than the NAFLD-CHB group

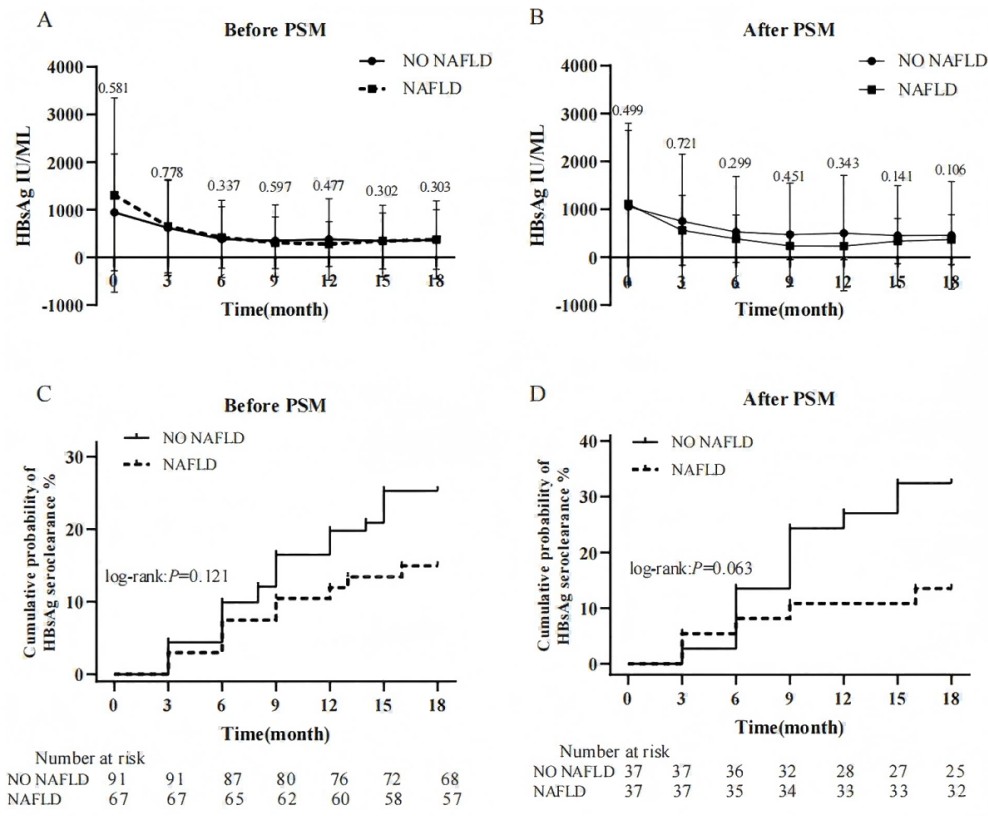

**Figure 3 Effect of NAFLD on HBsAg during PEG-IFN therapy.** (A) Changes in HBsAg levels between groups before PSM (B) Changes in HBsAg levels between groups after PSM (C) Cumulative probability of HBsAg seroclearance between groups before PSM. (D) Cumulative probability of HBsAg seroclearance between groups after PSM.

(25.3% *vs.* 14.9%), though this difference was not statistically significant (log-rank: $P = 0.121$). After PSM, the HBsAg clearance rates of the two groups were 32.4% and 13.5% respectively, and the difference was still insignificant ($P = 0.063$) (Figs. 3C and 3D). Additionally, median time to seroclearance did not differ significantly between groups before (9.0 *vs.* 7.5 months, $P = 0.750$) or after PSM (9.0 *vs.* 6.0 months, $P = 0.448$). Further analysis of the 33 patients who achieved HBsAg seroclearance revealed that 25 (75.8%) were anti-HBs positive, with a median level of 49.8 (IQR 10.8–215.2) mIU/ml. Among these, seven patients were in the NAFLD group. In the NAFLD group, three patients did not achieve HBsAg seroconversion. Comparison between the two groups showed no significant difference in seroconversion rates (30.0% *vs.* 21.7%, $P = 0.673$). After PSM, 17 patients achieved HBsAg seroclearance, of whom 14 were anti-HBs positive (10 in the CHB-only group). No statistically significant difference was observed between the groups ($P = 0.676$).

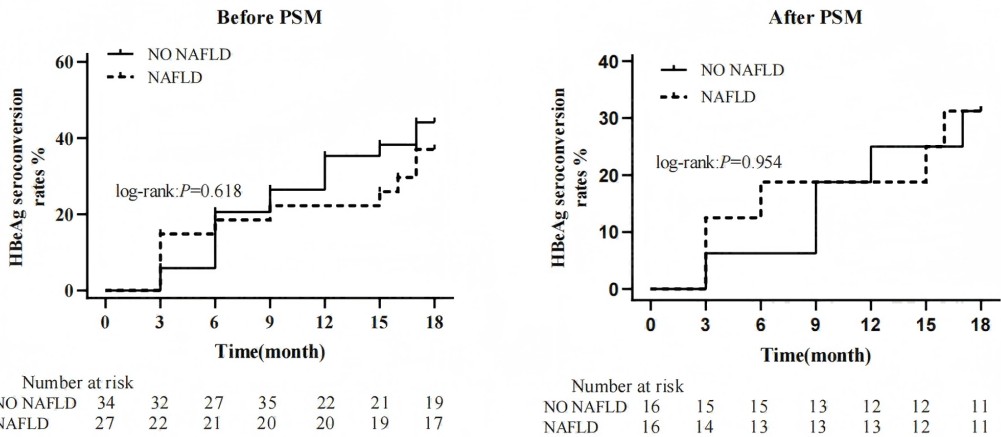

**Figure 4  Effect of NAFLD on HBeAg during PEG-IFN therapy.**

## The effect of NAFLD on HBeAg during PEG-IFN therapy in patients with CHB

At baseline, 61 patients (38.6%) were HBeAg-positive. Comparison of the cumulative incidence of HBeAg seroconversion at the end of follow-up revealed no significant difference between the two groups (log-rank: $P = 0.618$). After PSM, 32 patients (43.2%) remained HBeAg-positive. Similarly, no significant difference was observed in the cumulative incidence of HBeAg seroconversion between groups at the end of follow-up post-PSM (log-rank: $P = 0.954$) (Fig. 4).

## The analysis of factors associated with HBsAg seroclearance

In the univariate binary logistic regression analysis, lower baseline HBsAg level (OR 0.997; 95% CI [0.996–0.998]; $P < 0.001$), HBeAg-negative status (OR 2.560; 95% CI [1.033–6.344]; $P = 0.042$), higher total bilirubin (OR 1.065; 95% CI [1.010–1.124]; $P = 0.021$), and higher creatinine levels (OR 1.038; 95% CI [1.005–1.073]; $P = 0.024$) were associated with HBsAg seroclearance. Multivariate binary logistic regression confirmed that baseline HBsAg level (OR 0.997; 95% CI [0.995–0.997]; $P < 0.001$) and HBeAg-negative status (OR 1.246; 95% CI [1.025–3.531]; $P = 0.048$) were significantly associated with HBsAg seroclearance (Table 2).

Patients were stratified into four groups based on baseline HBeAg status and NAFLD diagnosis: Group 1: No NAFLD and HBeAg-negative Group; 2: No NAFLD and HBeAg-positive Group; 3: NAFLD with HBeAg-negative Group; 4:NAFLD with HBeAg-positive Group. Cumulative HBsAg seroclearance probabilities were compared across groups before and after PSM. No significant differences were observed among the four groups before PSM (32.8% *vs.* 12.1% *vs.* 17.5% *vs.* 7.4%, $P = 0.060$) and after PSM (42.9% *vs.* 18.8% *vs.* 18.2% *vs.* 6.7%, $P = 0.058$). Notably, regardless of HBeAg status, there was no conspicuous

**Table 2  Logistic regression of factors for HBsAg seroclearance.**

| | Univariate analysis | | Multivariate analysis | |
| --- | --- | --- | --- | --- |
| | OR (95% CI) | *P* | OR (95% CI) | *P* |
| SEX (Male) | 0.580 (0.186–1.808) | 0.348 | | |
| Family history | 0.525 (0.169–1.628) | 0.264 | | |
| NAs-treated time | 0.999 (0.895–1.116) | 0.988 | | |
| AGE | 0.998 (0.949–1.049) | 0.923 | | |
| Baseline HBsAg | 0.997 (0.996–0.998) | <0.001 | 0.997 (0.995–0.997) | <0.001 |
| Baseline HBeAg(-) | 2.560 (1.033–6.344) | 0.042 | 1.246 (1.025–3.531) | 0.048 |
| Baseline HBeAg | 0.896 (0.802–1.002) | 0.053 | | |
| ALT, U/L | 0.986 (0.964–1.008) | 0.201 | | |
| AST, U/L | 0.996 (0.963–1.029) | 0.798 | | |
| GGT, U/L | 0.997 (0.976–1.017) | 0.743 | | |
| ALP, U/L | 0.989(0.968–1.009) | 0.274 | | |
| AFP, ng/mL | 0.900 (0.733–1.106) | 0.317 | | |
| TP, g/L | 1.080 (0.979–1.192) | 0.124 | | |
| ALB, g/L | 1.103 (0.949–1.283) | 0.201 | | |
| TBIL, $\mu$mol/L | 1.065 (1.010–1.124) | 0.021 | | |
| DBIL, $\mu$mol/L | 1.236 (0.990–1.543) | 0.061 | | |
| UREA, mmol/L | 0.873 (0.599–1.271) | 0.478 | | |
| CREA, $\mu$mol/L | 1.038 (1.005–1.073) | 0.024 | | |
| UA, $\mu$mol/L | 0.999 (0.993–1.004) | 0.618 | | |
| GLU, mmol/L | 0.756 (0.394–1.453) | 0.402 | | |
| WBC, $\times 10^9$/L | 0.881 (0.690–1.125) | 0.310 | | |
| NLR | 1.120 (0.772–1.625) | 0.550 | | |
| PLT, $\times 10^9$/L | 0.997 (0.990–1.005) | 0.468 | | |

**Notes.**

ALT, alanine aminotransferase; AST, aspartate aminotransferase; GGT, gamma-glutamyl transferase; ALP, alkaline phosphatase; TP, total protein; ALB, albumin; TBIL, total bilirubin; DBIL, direct bilirubin; AFP, alpha fetoprotein; UREA, urea; CREA, creatinine; UA, Uric Acid; GLU, blood glucose; WBC, white blood cell; NLR, neutrophil-to-lymphocyte ratio; PLT, platelet.

difference in cumulative HBsAg clearance rates between the non-NAFLD and NAFLD groups before and after PSM ($P > 0.05$) (Fig. 5).

## The comparison of biochemical indices after PEG-IFN treatment between the two groups

Following PEG-IFN therapy, both groups exhibited remarkable elevations in ALT and GGT levels from baseline, peaking at weeks 12–24 before gradually declining. This study found that at each follow-up point, the NAFLD-CHB group exhibited substantially higher ALT and GGT levels *versus* the CHB-only group, with statistically significant intergroup differences (all $P < 0.05$; Fig. 6). During PEG-IFN treatment, 129 patients (81.6%) developed ALT elevations (>1 upper limit of normal (ULN)), of whom 108 achieved normalization by end-of-follow-up. The NAFLD-CHB group showed lower ALT normalization rates than the CHB-only group at week 24 (31.9% *vs.* 12.3%, $P = 0.009$) and week 48 (70.8% *vs.* 47.4%, $P = 0.007$). However, 24 weeks after the end of treatment, there

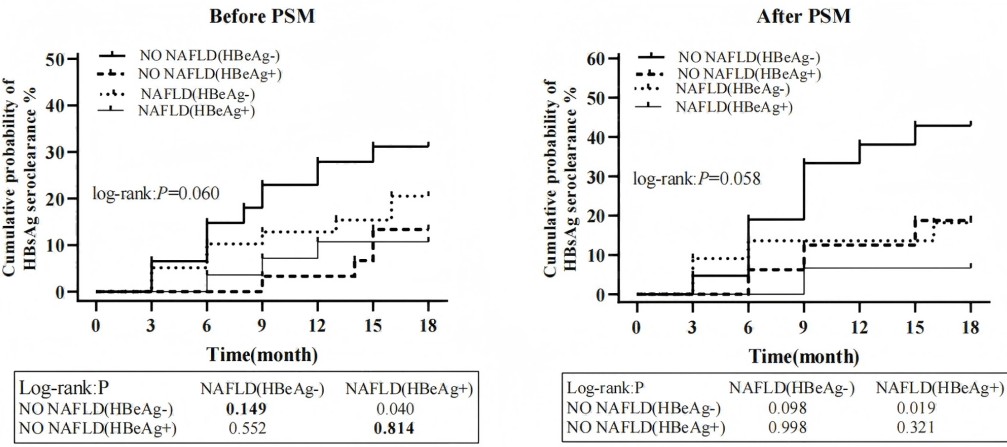

**Figure 5** The cumulative probability of HBsAg seroclearance stratified by baseline NAFLD and HBeAg (+/-).

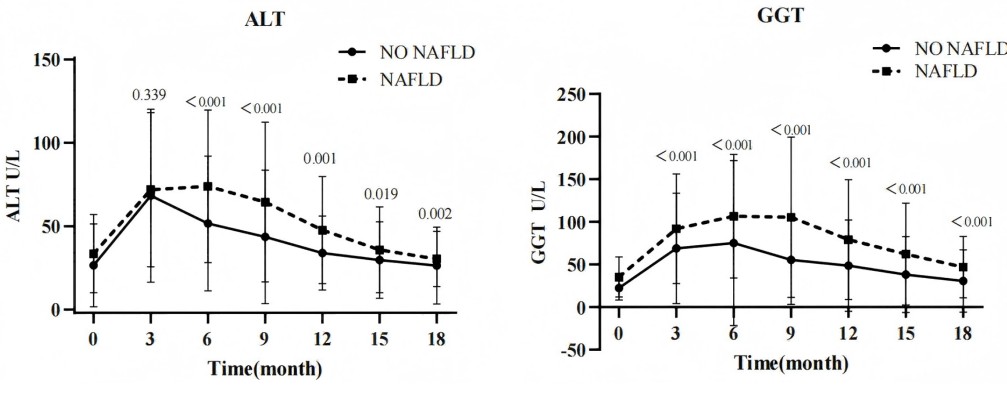

**Figure 6** Change trend of biochemical parameters.

**Table 3** Biochemical responses of the two groups with PEG-IFN treatment.

| Factors | CHB without NAFLD | CHB with NAFLD | $X^2$ | P |
|---|---|---|---|---|
| 24 week ALT normalization (n,%) | 23/72 (31.94) | 7/57 (12.28) | 6.892 | 0.009 |
| 48 week ALT normalization (n,%) | 51/72 (70.83) | 27/57 (47.37) | 7.328 | 0.007 |
| 72 week ALT normalization (n,%) | 64/72 (88.90) | 44/57 (77.19) | 3.193 | 0.074 |

was no significant difference in ALT normalization rates between the two groups (88.9% vs 77.2%, $P = 0.074$) (Table 3).

## The comparison of adverse events between the two groups

During treatment, 100 patients (63.3%) appeared adverse reactions. There was no significant difference in the incidence of adverse reactions between the CHB group and the NAFLD-CHB group (60.4% vs 67.2%, $P = 0.386$). Influenza-like symptoms (62.6%

**Table 4   Occurrence of adverse events.**

| Adverse effects | CHB without NAFLD ($n = 91$) | CHB with NAFLD ($n = 67$) |
|---|---|---|
| NO | 36 (39.6%) | 22 (32.8%) |
| Flu-like syndrome | 54 (59.3%) | 40 (59.7%) |
| Gastrointestinal symptoms | 16 (17.6%) | 18 (26.9%) |
| Myelosuppression | 44 (48.3%) | 27 (40.3%) |
| Skin allergy symptom | 3 (3.3%) | 4 (6.0%) |
| Neuropsychiatric symptom | 18 (19.8%) | 11 (16.4%) |
| Endocrine and metabolic diseases | 2 (2.2%) | 0 |

*vs.* 59.7%, $P = 0.964$) and hematological abnormalities (48.3% *vs.* 40.3%, $P = 0.315$) were most common, though without significant intergroup differences. Additionally, 34 patients (21.5%) reported decreased appetite with weight loss, showing no statistical significance between groups (40.3% *vs.* 26.9%, $P = 0.161$). Among them, 18 NAFLD-CHB patients exhibited weight reduction, yet follow-up imaging revealed no substantial improvement in hepatic steatosis. All enrolled patients' adverse reactions were alleviated through adjusted PEG-IFN dosages or symptomatic treatment (Table 4).

## DISCUSSION

In recent years, there has been considerable controversy in clinical practice and research regarding the effect of antiviral therapy in CHB patients with NAFLD, which has made it a focal research area globally. This retrospective study analyzed the impact of concurrent NAFLD on functional cure in NAs-treated CHB patients undergoing PEG-IFN antiviral therapy. Results demonstrated no significant difference in HBsAg seroclearance between NAFLD and non-NAFLD groups after PEG-IFN treatment, indicating that NAFLD does not compromise PEG-IFN's efficacy in achieving functional cure. Furthermore, during treatment, NAFLD-CHB patients exhibited significantly lower ALT normalization rates *versus* non-NAFLD counterparts; however, 24 weeks post-treatment, there was no significant difference in ALT normalization rates between groups.

Nevertheless, some studies indicate that NAFLD may facilitate HBsAg seroclearance in chronic HBV infections. Chu et al. reported that HBsAg carriers with moderate to severe hepatic steatosis exhibited 3.2- to 3.9-fold higher seroclearance rates compared to those without steatosis. They further observed that steatosis concurrent with elevated body mass index potentially facilitate HBsAg clearance in CHB patients (*Chu, Lin & Liaw, 2007*). These findings were further confirmed in their other cohort study (*Chu, Lin & Liaw, 2013*). Recently, *Mak et al. (2020)* reported that CHB patients with hepatic steatosis demonstrated higher HBsAg seroclearance rates than those non-steatotic. They proposed that intrahepatocytic lipid accumulation and associated metabolic dysregulation may alter subcellular HBsAg distribution, thereby inducing hepatocyte apoptosis, suppressing viral replication, and ultimately reducing HBsAg expression (*Huang et al., 2024*). It is worth noting that the subjects of the mentioned studies were mostly untreated CHB patients. The impact of concurrent NAFLD on the effectiveness of antiviral therapy for CHB

remains unclear. Our findings indicate that NAFLD does not significantly compromise interferon efficacy in NAs-treated CHB patients. Further mechanistic investigations are warranted. Our results align with prior studies, such as those by *Shi et al. (2012)* who found no significant impact of hepatic steatosis on the antiviral efficacy of PEG-IFN α-2a in CHB patients (*Cindoruk, Karakan & Unal, 2007*). A meta-analysis similarly demonstrated comparable HBeAg seroconversion and HBsAg seroclearance rates between steatotic and non-steatotic CHB patients, no matter what treated with NAs or PEG-IFN α (*Liu et al., 2023*). Therefore, regardless of whether hepatic steatosis affects HBV replication levels or HBsAg clearance in HBV infections, the consensus in the literature is that hepatic steatosis does not appear to affect the antiviral efficacy of interferon therapy in CHB patients.

This study demonstrated that baseline HBeAg status is a predictor of HBsAg seroclearance during interferon therapy, with significantly higher rates in HBeAg-negative *versus* HBeAg-positive patients. These findings are consistent with other research in the field (*Yeo et al., 2019*). Crucially, irrespective of HBeAg status, no significant difference in HBsAg seroclearance emerged between non-NAFLD and NAFLD groups. This further confirms that concurrent NAFLD does not have a pronounced impact on achieving clinical cure in NAs-treated CHB patients with pegylated interferon therapy.

In this study, ALT and GGT levels increased substantially from baseline in both groups following PEG-IFN therapy, peaking between weeks 12–24 before gradual decline. This transaminase elevation represents a favorable predictor of immune reconstitution during interferon therapy, potentially mediating virological response through enhanced natural killer (NK) and T-cell activity (*Ghany et al., 2020*). We observed that ALT and GGT levels were higher in the NAFLD group compared to the NO-NAFLD group at multiple follow-up points. This elevation likely stems from hepatocyte lipid accumulation-induced cellular damage, promoting enzyme leakage into circulation (*Zhang et al., 2023*). Furthermore, a meta-analysis confirmed significantly lower ALT normalization rates in NAFLD-CHB patients during short-term antiviral therapy, with rates becoming comparable after extended follow-up (*Rui et al., 2024*). This aligns with our findings. This may be related to the steatosis-driven inflammatory responses that perturb liver biochemistry and attenuate early antiviral treatment benefits (*Rui et al., 2024*).

Common adverse events during interferon therapy encompass bone marrow suppression, flu-like symptoms (*e.g.*, fever, myalgia, headaches, fatigue), and neurological symptoms (including somnolence and depression). Most reactions are mild and reversible (*Zhang et al., 2024*). In this study, NAFLD was found to have no significant effect on adverse events in CHB patients treated with PEG-IFN. Notably, this study did not find any remission or progression of fatty liver in CHB patients with concurrent NAFLD during interferon therapy. Further research is needed to clarify the impact of interferon therapy on hepatic steatosis.

This study has several limitations. Firstly, as a single-center retrospective investigation, this research carries potential methodological biases inherent to its design. To systematically control these risks, we implemented the following rigorous measures: During the data collection phase, detailed and stringent inclusion/exclusion criteria were established. In the analytical phase, PSM was employed to balance intergroup disparities. So as to

minimize the interference of bias on the research results as much as possible. Secondly, most clinical data for this study were extracted from outpatient electronic medical records, lacking comprehensive anthropometric measurements (*e.g.*, height, weight) and blood lipid profiles, which make it difficult to analyze factors related to hepatic steatosis. Thirdly, the diagnosis of NAFLD in this study lacked histological validation, precluding assessment of steatosis severity's impact on interferon efficacy. Therefore, multicenter prospective studies with extended follow-up periods and precise steatosis quantification methods are needed to confirm the impact of hepatic steatosis on antiviral therapy in CHB patients.

## CONCLUSIONS

In conclusion, this study demonstrated that NAFLD does not significantly affect the functional cure of nucleoside-treated CHB patients receiving PEG-IFN therapy. Moreover, NAFLD-CHB patients exhibited lower ALT normalization rates than CHB-only counterparts at the end of treatment, though this difference became non-significant with extended follow-up. The ultimate impact of NAFLD on liver inflammation progression requires long-term observation for clarification. Therefore, for CHB patients with comorbid NAFLD, while actively administering antiviral therapy, it is recommended to establish a standardized and dynamic fatty liver monitoring mechanism. This mechanism should closely track the progression or regression of fatty liver through regular imaging assessments and metabolic indicator testing, enabling timely optimization of clinical intervention strategies.

## ACKNOWLEDGEMENTS

Thanks to all colleagues who contributed to this manuscript, and thanks to the Xuzhou Medical University Hospital for providing patient data.

### Funding

The authors received no funding for this work.

### Competing Interests

The authors declare there are no competing interests.

### Author Contributions

- Huili Li conceived and designed the experiments, performed the experiments, analyzed the data, prepared figures and/or tables, authored or reviewed drafts of the article, and approved the final draft.
- Ling Li performed the experiments, prepared figures and/or tables, and approved the final draft.
- Yiru Zhao performed the experiments, prepared figures and/or tables, and approved the final draft.

- Guangde Yang analyzed the data, prepared figures and/or tables, and approved the final draft.
- Xia Wang analyzed the data, prepared figures and/or tables, and approved the final draft.
- Juanjuan Fu analyzed the data, authored or reviewed drafts of the article, and approved the final draft.
- Li Li analyzed the data, authored or reviewed drafts of the article, and approved the final draft.
- Xiucheng Pan conceived and designed the experiments, authored or reviewed drafts of the article, and approved the final draft.

## Human Ethics

The following information was supplied relating to ethical approvals (i.e., approving body and any reference numbers):

The ethics committee of Xuzhou Medical University Affiliated Hospital approved the study (Ethics number: XYFY2024-KL225-01).

## Data Availability

The raw data is available in the Supplemental Files.

## Supplemental Information

Supplemental information for this article can be found online at http://dx.doi.org/10.7717/peerj.19972#supplemental-information.

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
