# Peer review of "Impact of nonalcoholic fatty liver disease on the functional cure of nucleos(t)ide analogues-treated chronic hepatitis B patients add-on pegylated interferon therapy: a retrospective study"

_PeerJ, doi:10.7717/peerj.19972_

## Round 0.1 · original submission · Major Revisions

**Language Note:** The review process has identified that the English language must be improved. PeerJ can provide language editing services - please contact us at [email protected] for pricing (be sure to provide your manuscript number and title). Alternatively, you should make your own arrangements to improve the language quality and provide details in your response letter. – PeerJ Staff

Reviewer 1 ·

Basic reporting

-

Experimental design

1. The retrospective nature limits causality and introduces potential biases related to patient selection and data collection. Prospective studies are necessary to confirm the findings. The authors should clearly address the limitations of the retrospective design and discuss how they sought to mitigate potential biases.
2. The sample size of 158 patients may restrict the statistical power and generalizability of the study findings.
3. The study's definition and diagnostic criteria for MAFLD require a more detailed explanation and justification for the chosen approach based on the latest APASL 2025 guidelines (PMID: 40016576).
4. The influence of confounding factors (apart from MAFLD) on the outcomes requires more rigorous examination and adjustment.
5. While PSM was applied, additional sensitivity analyses were needed to evaluate the robustness of the findings in different subgroups.
6. Longer follow-up periods are essential to assess the long-term effects of MAFLD on treatment outcomes and disease progression.
7. A more detailed analysis of the types and severity of adverse events and their relationship to MAFLD would be beneficial.
8. Further subgroup analyses (e.g., based on BMI or liver fibrosis stage), stratified by various factors, are essential for identifying specific patient populations that may benefit the most from the treatment.

Validity of the findings

-

Reviewer 2 ·

Basic reporting

A professional English speaker must approve this article.

Experimental design

-

Validity of the findings

Please make a conclusion and a suggestion based on the study's findings. This study didn't mention dietary habits and lifestyle. The suggestion should be relevant to the conclusion.

Additional comments

"After PSM, the HBsAg clearance rates of the two groups were 32.4% and 13.5%,
(P = 0.063)"
If we look at the absolute percentage rate, the difference is nearly 20%, but the P value is not significant. In a clinical setting, this difference in HBsAg clearance is very important. Is there any possible explanation for this finding?

Reviewer 3 ·

Basic reporting

-

Experimental design

-

Validity of the findings

-

Additional comments

1. Not all measurement data were non-normal continuous distributions. The continuous data with normal distributions should be presented as mean with SD, and analyzed by t-test.
2. If the data are too small, the categorical data should be compared using Fisher’s exact test.
3. Multivariate analysis should be applied to Table 2.
4. Propensity score matching (PSM) is a good method to overcome baseline imbalances. However, the patient number will become too small. The authors may try the method of Inverse probability of treatment weighting (IPTW).
5. The English needs polishing.
Although the priority of this study is not so high, this manuscript provides useful information for clinicians to manage chronic hepatitis B patients.

---

## Round 0.2 · accepted · Accept

All issues pointed out by the reviewers were adequately addressed, and the revised manuscript is acceptable now.

Reviewer 1 ·

Basic reporting

I appreciate the author's thorough response to all my comments. I have no further comments.

Experimental design

Good

Validity of the findings

Good

Additional comments

I appreciate the author's thorough response to all my comments. I have no further comments.

Reviewer 3 ·

Basic reporting

No comment

Experimental design

No comment

Validity of the findings

No comment

Additional comments

The authors have responded the comments and suggestions appropriately.